# Disentangling LiDAR Contribution in Modelling Species–Habitat Structure Relationships in Terrestrial Ecosystems Worldwide. A Systematic Review and Future Directions

**Pablo Acebes** [1,2,*], **Paula Lillo** [1] **and Carlos Jaime-González** [1]

[1] Departamento de Ecología, Facultad de Ciencias, Universidad Autónoma de Madrid, 28049 Madrid, Spain; paulalilloaparici@gmail.com (P.L.); carlos.jaime.g@gmail.com (C.J.-G.)

[2] Centro de Investigación en Biodiversidad y Cambio Global (CIBC-UAM), Universidad Autónoma de Madrid, 28049 Madrid, Spain

\* Correspondence: pablo.acebes@uam.es

**Abstract:** Global biodiversity is threatened by unprecedented and increasing anthropogenic pressures, including habitat loss and fragmentation. LiDAR can become a decisive technology by providing accurate information about the linkages between biodiversity and ecosystem structure. Here, we review the current use of LiDAR metrics in ecological studies regarding birds, mammals, reptiles, amphibians, invertebrates, bryophytes, lichens, and fungi (BLF). We quantify the types of research (ecosystem and LiDAR sources) and describe the LiDAR platforms and data that are currently available. We also categorize and harmonize LiDAR metrics into five LiDAR morphological traits (canopy cover, height and vertical distribution, understory and shrubland, and topographic traits) and quantify their current use and effectiveness across taxonomic groups and ecosystems. The literature review returned 173 papers that met our criteria. Europe and North America held most of the studies, and birds were the most studied group, whereas temperate forest was by far the most represented ecosystem. Globally, canopy height was the most used LiDAR trait, especially in forest ecosystems, whereas canopy cover and terrain topography traits performed better in those ecosystems where they were mapped. Understory structure and shrubland traits together with terrain topography showed high effectiveness for less studied groups such as BLF and invertebrates and in open landscapes. Our results show how LiDAR technology has greatly contributed to habitat mapping, including organisms poorly studied until recently, such as BLF. Finally, we discuss the forthcoming opportunities for biodiversity mapping with different LiDAR platforms in combination with spectral information. We advocate (i) for the integration of spaceborne LiDAR data with the already available airborne (airplane, drones) and terrestrial technology, and (ii) the coupling of it with multispectral/hyperspectral information, which will allow for the exploration and analyses of new species and ecosystems.

**Keywords:** biodiversity and habitat mapping; conservation; ecosystem structure; LiDAR platforms; LiDAR traits; morphological traits; remote sensing; terrain topography

## 1. Introduction

Global biodiversity is threatened by unprecedented and increasing anthropogenic pressures, including habitat loss and fragmentation, so that biodiversity assessment and monitoring is imperative [1,2]. Given that ecosystem structure is increasingly seen as a determinant of habitat quality as well as an indicator of biodiversity itself at local and regional scales [3–5], the ability to deepen our knowledge on species–habitat relationships is of vital importance [6].

The most direct and accurate way of obtaining detailed ecosystem three-dimensional (3D) structure at the resolution and accuracy required is through LiDAR [7,8]. Equipped

with a near-infrared light sensor in most cases and a GPS, Light Detection and Ranging (LiDAR is an active remote sensing technology able to measure the return time of an emitted laser pulse to obtain the distance to the object of study and thus its location in 3D (x, y, and z). Once the laser return is taken, a so-called "point cloud" is generated, from which the 3D model is constructed [7].

LiDAR sensors can be mounted on several platforms, thus providing different properties for researchers. For example, spaceborne (satellite) platforms commonly provide open access data to a broad, spatial extent, albeit at a coarse spatial resolution (e.g., [9,10]), promising opportunities to come from recently launched missions [11]. Platforms onboard airplanes or helicopters (Airborne Laser Scanning, or ALS) are frequently used because country-wide datasets are becoming increasingly available, offering increased accuracy, but at higher costs (e.g., [12,13]). Unmanned Aerial Systems (also called drones; hereinafter, UAS) are gaining relevance—their capacity for flying slow and above the vegetation or the ground provides a very high spatial resolution and at a lower cost on local scales [14]. Finally, ground-based instruments, commonly mounted on tripods or on handheld devices, also called Terrestrial Laser Scanning (TLS), offer the highest spatial resolution but a limited spatial coverage [15]. Thus, choosing the best option is not always an easy task for researchers, even more so for nonspecialized practitioners.

LiDAR technology has been largely used in forestry to measure 3D forest structure [16]. More recently, LiDAR has been incorporated into ecological studies for exploring, explaining, and predicting biodiversity at different scales [8,17,18]. Such studies typically use LiDAR-based metrics of vegetation canopy structure, as canopy height, cover, and vertical distribution [7,17], given that vegetation structure shapes niche variability and provides essential elements for species' habitats [3,19]. Most of these studies are focused on birds, due to their strong dependence on vegetation structure [3], as seen in recent reviews indicating LiDAR's outstanding performance [17,18]. However, new studies have pointed to the ability of LiDAR-derived metrics to model species–habitat relationships of other taxonomic groups, such as mammals [20,21], arthropods [22,23], reptiles and amphibians [24,25], or even bryophytes, lichens, and fungi [26,27]. Yet, there is no global overview available for the contribution of LiDAR metrics considering such diverse organisms. In addition, studies commonly neglect to discuss the most appropriate and accurate LiDAR metrics among the available set. This is particularly relevant because vegetation metrics can be estimated in many ways, so that comparing model outcomes becomes challenging [18]. Hence, harmonizing such diverse metrics in a few morphological (structural) traits is recommended. Something similar happened with non-forest habitats, wherein new studies have suggested remarkable LiDAR performance in modelling habitat attributes for species in savannah, tundra, or desert ecosystems [28–30]. In these open-landscapes, other elements such as shrubs or precise topography derived from fine-scale LiDAR data may be decisive for modelling species–habitat relationships, even more so than vegetation structure.

Here, we review the use of LiDAR to analyze 3D ecosystem structure (vegetation physiognomy and topography) in relation to the abundance, occurrence, richness, and diversity of species, including behavioral studies in terrestrial ecosystems. Firstly, we summarize the different taxonomic groups for which LiDAR has been used to model ecosystem features, their geographical coverage, and the ecosystem represented. We consider birds, mammals, reptiles, amphibians, and invertebrates. We also include bryophytes, lichens, and fungi, since these broad groups are becoming widely considered in biodiversity conservation and global change research, so that LiDAR technology may provide further valuable information [26,27]. We then extract LiDAR-derived metrics and harmonize them into five structural traits aiming to simplify the multiple metrics currently used, and to get an overview of which are the most widespread. Further, we analyze the most influential (significant) traits for each taxonomic group and ecosystem. We also examine whether LiDAR data was used alone or in combination with other resources (passive remote sensing, field data, and/or GIS-derived products). We then review some LiDAR technical attributes and characteristics, such as the type of platform on which LiDAR sensors are mounted

(spaceborne, airborne, or ground-based instruments), the point cloud spatial resolution, the type of LiDAR signal, and the availability of LiDAR data. Finally, we discuss and try to shed light on new opportunities forthcoming from UAS LiDAR and spaceborne missions, as well as from the combination of structural and spectral information aiming at describing fine-scale environmental heterogeneity for biodiversity mapping and monitoring.

## 2. Materials and Methods

We performed a systematic literature search using the ISI Web of Knowledge (WOK) and Scopus online databases to identify articles linking LiDAR technology to biodiversity patterns during the period 2000–2020 (2000 was the first year where we found an article that fulfilled our queries). We focused our bibliographic queries on vertebrate (birds, mammals, reptiles, and amphibians) and invertebrate fauna (only species belonging to arthropod and mollusk phyla were registered), and on bryophytes, lichens, and fungi. To do so, we used the following keyword strings, looking for matches in the title, abstract and/or keywords: "LiDAR* AND biodiversity*", "LiDAR* AND vertebrates*", "LiDAR* AND mammals*", "LiDAR* AND birds*", "LiDAR* AND reptiles*", "LiDAR* AND amphibians*", "LiDAR* AND invertebrates*", "LiDAR* AND insects*", "LiDAR* AND fungi*", "LiDAR* AND lichens*", "LiDAR* AND bryophytes*".

From the initial search (updated on 15 June 2020), we retrieved 3517 records (WOK: 2816, Scopus: 701; Supplementary Materials Table S1). We selected every record where LiDAR was used to measure and relate 3D ecosystem structure (vegetation physiognomy and terrain topography) to abundance, occurrence, richness, and diversity of species, including behavioral studies. Articles not related to the ecology field and those using LiDAR to measure vegetation structure for forestry (and hydrology) research (i.e., not relating vegetation structure or topography to any organism) were discarded, as well as methodological papers and reviews. In a second selection, all duplicated records and conference papers were removed. The finally selected records included 173 papers that fulfilled our search criteria (the complete list of references is available in the Supplementary Materials section).

We extracted the following information from each article: title, author(s), year of publication and journal. We also recorded the studied taxonomic group, including order, family, and the species when available (for a complete list of the species recorded, see Supplementary Materials, Table S2). We then classified them into five broad groups: 'Birds and flying mammals (Chiroptera)', hereinafter 'Birds', due to their similar use of space; 'Mammals (except Chiroptera)', hereinafter 'Mammals'; 'Reptiles and amphibians'; 'Invertebrates', which includes insects, arachnids, mollusks, and crustaceans (no other invertebrate groups were recorded in the literature search); 'Bryophytes, lichens and fungi' (hereinafter BLF).

We also recorded the study location, which was assigned to a continent to know the global coverage of studies (Africa, North America, Central–South America, Asia, Europe, and Oceania), and the type of ecosystem where the study was conducted. Since ecosystems were varied (e.g., Mediterranean coniferous and deciduous forests, evergreen and deciduous temperate forests), we harmonized them into the following broad ecosystems to simplify their nomenclature: temperate forest, rainforest, riparian forest (temperate floodplain rivers), taiga, tundra, savannah, arid and semiarid ecosystems, inland and coastal waters, and agroecosystem. The latter included arable and permanent land, small-wooded patches within an agricultural matrix, and grazed pasture.

We extracted all LiDAR metrics used to explain ecological patterns. Since the use of metrics and its naming were diverse, we harmonized them into the following morphological (structural) traits: (a) Canopy structure (that we divided into canopy height, canopy cover and canopy vertical distribution); (b) Understory structure and shrubland; (c) Topography (Table 1). We then computed these traits as the number (and percentage) of studies that used, e.g., canopy cover traits for the whole set of articles, and for each taxonomic group and ecosystem analyzed. As an indicator of each traits' performance,

we further identified which ones were significant in the modelling exercises, since their explanatory power may differ among the species and ecosystems. This index was estimated as the percentage of articles where the traits were identified as significant. In some cases, it was difficult to unequivocally identify whether a given metric was influential, for example, because it was significant only under particular circumstances. In those cases, we considered the traits as influential if it had an effect.

We registered LiDAR spatial resolution through the point cloud density (number of points/m$^2$); while high point density ($\geq$10 points/m$^2$) allowed us to obtain detailed information of canopy layers [31,32], low point density (<2 points/m$^2$) was enough to create Digital Terrain Models (DTMs), or to calculate canopy height and cover metrics [33,34]. We also recorded the type of signal, distinguishing between Discrete Return (DR) and Full Waveform (FW). DR are small footprint systems that record one to several echoes, but only if the echo exceeds a predefined threshold of intensity. As a result, echoes not strong enough can be missed, e.g., echoes coming from single tree branches. Contrarily, FW provides a continuous distribution of laser energy for each pulse, regardless of the echo strength, but data processing is more complex, and publicly available data is less common [35].

**Table 1.** LiDAR traits used to harmonize and classify the whole set of LiDAR metrics included in the reviewed papers (173), their meaning, and some examples. CS: canopy structure.

| LiDAR Traits | Description | Examples of LiDAR Explanatory Variables |
|---|---|---|
| CS: canopy height | Includes metrics used to obtain canopy height measures (e.g., maximum, mode, 95th percentile, etc.). | Mean height [36,37]; Mean outer canopy height [38]; 25th and 95th percentiles [39,40]; Maximum height [37,41]. |
| CS: canopy cover | Includes metrics describing canopy horizontal structure (e.g., cover, gaps, density, roughness, etc.). | Canopy cover: Number of first returns above a height/total number of first returns [42]; Intensity sum of all non-ground points divided by intensity sum of all points [43]; Canopy gaps and ruggedness [44]; Fractional cover [45]; LiDAR penetration ratio (LPI), as the ratio between terrain points and total points [22]. |
| CS: canopy vertical distribution | Includes metrics describing canopy vertical variation (heterogeneity, complexity, layering, etc.). | Standard deviation of vegetation height [46]; Proportion of LiDAR returns at different layers [47]; Gini coefficient [48]; Simpson index [39]; Vertical gap index, measured as the total distance between individual canopy strata divided by maximum canopy height [43]. |
| Understory structure and shrubland | Describes height, cover, volume and/or contribution of the strata below tree canopy and shrub structure in non-forest habitats. | Index of foliage height diversity, such as the Shannon diversity index for the returns between 0.5 m and 3 m [49]; Density of understory cover: ratio of understory returns to the total number of understory and ground returns [33]; Shrub and snags cover: vegetation returns between 1 and 2.5 m [50]; Understory penetration ratio [36]. |
| Topography | GIS variables commonly derived from LiDAR data for describing surface topography (elevation, slope, aspect, terrain roughness) and derived variables such as soil moisture, solar radiation, etc. | Rugosity [51]; Slope, aspect, curvature [52]; Topographic wetness index [30,53]; Depth-to-water Index [26]; Potential incoming solar radiation [30]. |

We recorded whether LiDAR data was public and freely available (e.g., LiDAR data from National forestry inventories, commonly of lower spatial resolution [41,54], or either access-restricted or privately conducted ad hoc [40,55]. Public availability may be an indicator of the democratization of LiDAR use.

We registered the type of LiDAR platform, since each platform provides different coverage and spatial resolution: spaceborne (satellite), airborne (plane, helicopter or UAS), and TLS. We also recorded whether LiDAR data was used from one or several periods, either leaf-on leaf-off data of a single period [12] or multi-annual data [56] to identify potential ecosystem structural changes and their effects on species.

Finally, we registered whether LiDAR data was used alone or in combination with other resources, which we classified as active or passive remote sensing imagery, field-based studies, and GIS-derived products. Moreover, we registered whether they were used to compare LiDAR performance or as a complement, since this information gives an idea of how confident ecologists are in LiDAR technology.

## 3. Results

Our search retrieved 173 articles published from 2000 to 2020 with relevant information linking LiDAR technology to biodiversity patterns for the selected taxonomic groups. We did not record any paper before 2000 with the specific keyword strings used. There was an increasing trend in the number of articles published over time (Figure 1).

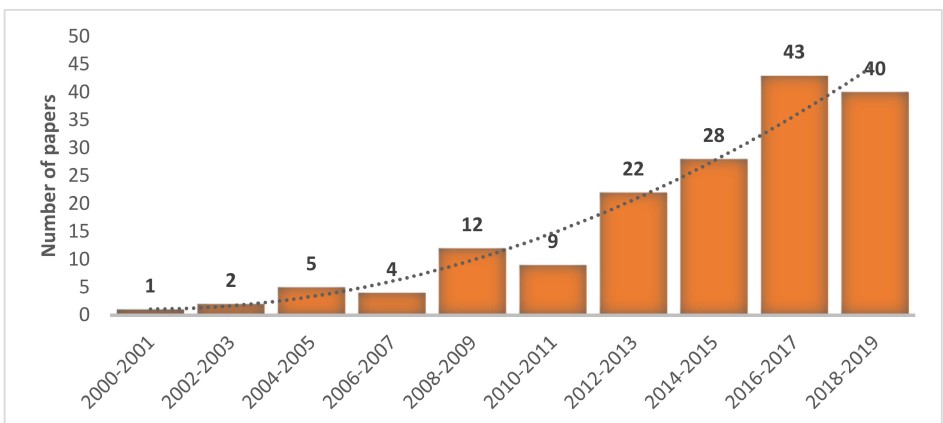

**Figure 1.** Temporal trends in the number of published articles linking LiDAR-based variables to the studied taxonomic groups for the period 2000–2019: Birds, mammals, reptiles and amphibians, invertebrates, bryophytes, lichens, and fungi. Note that no article published before 2000 was retrieved using search strings (see methods). The year 2020 was not included, since literature search was conducted until June.

### 3.1. Geographic Distribution, Taxonomic Groups and Ecosystems Surveyed

There was still a remarkable difference in the number of articles published in the Northern Hemisphere, specifically in Europe and North America; no studies were consistently conducted outside these two continents until 2015. Only two papers had a worldwide focus or were performed in more than one continent (Figure 2).

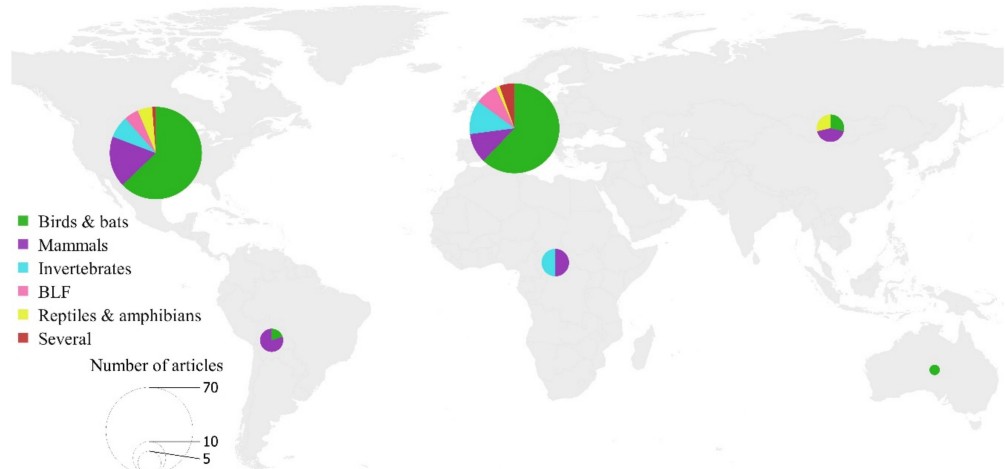

**Figure 2.** Number of articles per continent (size of the circle) found in the literature search (2000–2020) that relate to metrics based on LiDAR with the studied taxonomic groups and the proportion of articles of these taxonomic groups by continent. BLF: Bryophytes, lichens, and fungi; Several: studies including more than one taxonomic group. See main text for more details.

Birds were the most studied taxonomic group (N = 99, 57.2%), followed by mammals (N = 33, 19.1%), invertebrates (N = 18, 10.4%), and BLF (N = 10, 5.8%). Reptiles and amphibians were the least represented group (N = 7, 4%). BLF was the only group where studies have been conducted in North America and Europe, particularly in latitudes above 48°N (Alaska, Canada, Denmark, Finland, Germany, and Norway), except one study predicting biological crust in arid environments in southern Europe [30]. Six articles analyzed more than one taxonomic group (Figure 2).

There was a trend towards monospecific studies in birds and especially in mammals, while there was a pattern of multispecific studies in invertebrate and in BLF groups. Regarding birds, 59.6% of the articles (N = 59) analyzed either a single species or a few related species, while the remaining 40.4% (N = 40) studied bird richness and diversity patterns. Specifically, 29 articles focused on Passeriformes order, 11 on Piciformes, 8 on Galliformes, 7 on Strigiformes, and 2 on Charadriiformes, while 6 considered the Chiroptera species (bats). For mammals, only seven articles (21%) focused on more than one species, while the remaining 79% (N = 26) were monospecific studies. The best represented order was Artiodactyla (N = 11), followed by Rodentia (N = 8), Carnivora (N = 8), Primates (N = 5), and Lagomorpha (N = 2).

Regarding invertebrates, only three articles studied a single species (*Procambarus clarkii*, [57]; *Nysius wekiuicola*, [15]; *Crassostrea virginica*, [58]), while the remaining 15 were multispecific studies (83%). Five articles examined Coleoptera order, three Isoptera order, two Lepidoptera order, one Araneae order, one Hemiptera order, one Ostreida order, one the Mollusca phylum, and one the Crustacea sub-phylum. There was also one article considering several orders (Coleoptera, Hemiptera, Araneae, Hymenoptera, [59]), one article focusing on arthropod diversity [12], and one article analyzing benthonic invertebrate richness and abundance [23].

With regard to BLF studies (N = 10), six articles focused on bryophyte and/or lichen distribution, two articles analyzed fungi richness, and another two focused on fungi distribution; one article studied *Cladonia* and *Cetraria* lichens genera, while one article was centered on BLF diversity across multiple habitats to a large geographic extent [27].

For reptiles and amphibians, two articles focused on species belonging to Crocodilia and Squamata order respectively, and two on species belonging to Testudines order, while the remaining three analyzed potential breeding ponds for amphibians [25].

Finally, among the articles considering several taxonomic groups, two studies analyzed richness and abundance of birds and insects [49,60], two articles focused on bird, amphibian, and mammal richness [9,61], one studied the diversity of birds and butter-

flies [62], and one study analyzed bats, birds, invertebrates, fungi, lichens, and bryophytes diversity [36].

Temperate forest was by far the most represented ecosystem (N = 116, Figure 3A) and the most common across all taxonomic groups (Figure 3B). Contrarily, riparian forests (only in bird studies), savannah (mammals and invertebrates), and tundra ecosystems (birds and BLF) were the least represented (Figure 3B). Studies on birds and invertebrates showed the greatest ecosystem variety (eight and six ecosystems, respectively; Figure 3B), while the remaining taxonomic groups were studied in five different ecosystems. This included six mammal studies in rainforest [10,21,63–66], four studies in African savannahs [67–70], three articles of invertebrates in savannahs [28,71,72], or seven articles of birds in agroecosystems [53,73–78]. Whereas reptile studies were conducted in the rainforest [24], agroecosystem [79], or inland and coastal waters [51], amphibian studies were restricted to seasonal ponds within temperate forests (N = 3). Five studies corresponding to four taxonomic groups were conducted in arid and semiarid ecosystems: birds [80], BLF [30], reptiles [47], and invertebrates [15,57].

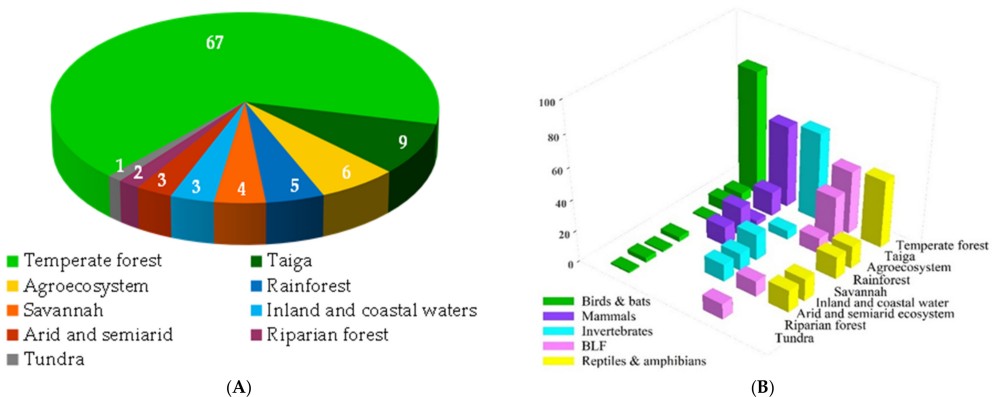

**Figure 3.** Percentage of articles found in the literature search dealing with LiDAR data and the studied taxonomic groups (**A**) grouped by the type of ecosystem where each study was conducted, and (**B**) arranged considering the type of ecosystem by taxonomic group. BLF: Bryophytes, lichens, and fungi; Several: studies including more than one taxonomic group. Note that the sum of percentage of studies by ecosystem within each taxonomic group is equal to 100%. See main text for more details.

### 3.2. LiDAR Traits: Use versus Performance

Canopy height (62.4%) and canopy cover (58.4%) were the most used LiDAR traits for the whole set of articles, followed by canopy vertical distribution (45.1%), understory structure and shrublands (33.5%), and topography (30.1%). Globally, canopy cover and topography traits performed best (93.2% and 86.5%, respectively), followed by canopy height (84.3%), understory structure and shrublands (81.0%), and canopy vertical distribution (74.4%).

By taxonomic group, canopy height was the most used LiDAR trait in bird and mammal studies, even though canopy cover was more influential (Table 2). Although relevant, canopy vertical distribution traits were less important for birds than canopy height and cover. These traits were also relevant for invertebrates, and, to a lesser extent, for mammals. Understory and shrublands traits were used less often than other vegetation structure traits in BLF and invertebrates, despite the fact they were influential for those taxa. Understory and shrubland traits were also relevant for mammals and birds. Terrain topography traits were the most used and influential in BLF, reptiles, amphibians, and invertebrates (Table 2).

Regarding the type of ecosystem, canopy height, together with canopy cover, were the most used traits in forest-type ecosystems (Table 3). In those ecosystems, canopy vertical distribution traits were widely used, though with less influence. In contrast, in more open ecosystems, such as in arid and semi-arid ecosystems, savannah, or inland and coastal waters, topographic traits were broadly used. No studies conducted in rainforests used

understory structure traits. In contrast, understory structure and shrubland traits were influential in those ecosystems in which they were used (Table 3).

### 3.3. LiDAR Characteristics

Airborne LiDAR was the most used platform (94.8%, N = 164), while DR was the most used signal type (N = 152); only 21 articles used a FW signal. Seven out of 164 articles mounted the LiDAR sensor on a helicopter instead of on an airplane [12,32,81–85], which increased LiDAR resolution because the helicopter's flight was lower and slower (helicopter's pulse density: 15–500 pulses/m$^2$). Only four studies used LiDAR data from space sensors, specifically the publicly available Geoscience Laser Altimeter System (GLAS) onboard the ICE, Cloud and land Elevation Satellite (ICESat), and available from 2003 to 2009. Four articles acquired LiDAR data from TLS: DTMs [15,86]; canopy structural metrics [38] and shrub canopy structure [80]. According to our query, we did not record any study using UAS (drones).

Regarding point cloud resolution (pulse density), LiDAR data had less than two points/m$^2$ in 28.9% of the studies (N = 50), and between two and ten points/m$^2$ were recorded in 24.3% (N = 42), whereas 15.6% of the articles had more than 10 points/m$^2$ in (N = 27). Point cloud resolution was either not available, not explicitly stated, or was expressed as point spacing (distance from point to point) in 31.8% of the articles (N = 55). Private data were the most common LiDAR source (N = 100; 57.8%), followed by freely accessible LiDAR (N = 43; 24.9%), while 20 articles used both private and public datasets (5.8%). It was not possible to know the origin of data in 10 articles (5.8%). Only nine of the 43 articles using freely accessible data had more than two points/m$^2$ of spatial resolution, pointing to a clear lower accuracy of freely distributed LiDAR data. It was not always easy to identify the data source throughout the studies.

### 3.4. Use of Multi-Temporal LiDAR and Combination of LiDAR Data with Other Sources

Only five articles used LiDAR data to quantify temporal changes. For example, [87] used a time series of LiDAR data in a 12 year period to investigate the accuracy of LiDAR metrics when modelling the breeding period of the great tit (*Parus major*) in a mature woodland. The authors of [12] analyzed two LiDAR datasets (winter and summer) within the same year to check the reliability of LiDAR to penetrate leaf-on (as opposed to leaf-off) vegetation. The authors of [88] suggested that a 6 year difference between field-data collection and LiDAR-data collection had negligible effects on bird patterns in an undisturbed coniferous forest.

Regarding the use of LiDAR with other sources of information, 67 studies used only LiDAR data, whereas the remaining 106 shared LiDAR with other data sources (Figure 4). Field surveys (N = 42) and passive remote sensing imagery (N = 38) were the most widely implemented sources, including one article using hyperspectral data. Thirty-eight studies used these approaches to compare the reliability of each data source, 65 used them to improve modelling exercises, while the remaining three compared and complemented LiDAR data. There was a clear trend since 2014–2015 towards a decrease in the number of articles comparing methods and an increase in the number of studies that complemented LiDAR with other technologies.

**Table 2.** Categorized LiDAR-derived traits based on all metrics found in the literature search regarding vegetation structure and terrain topography in relation to the studied taxonomic groups. "n" represents the number of studies of each taxonomic group in which a variable regarding the categorized traits was used; "use" is the percentage of articles of each taxonomic group that uses a given metric in relation to the whole articles of each taxonomic group. For example, if 68 of 99 of bird studies used canopy height, then the use was 68.7%. "sig." is the ratio (%) between the number of articles in which a given metric has an effect in relation to the number of articles in which that metric is used. For example, in 59 of 68 bird articles in which canopy height was used, it had an effect = 86.76%. BLF: Bryophytes, lichens, and fungi; R&A: reptiles and amphibians; Several: studies including more than one taxonomic group.

| | Canopy Height | | | Canopy Cover | | | Canopy Vertical Distribution | | | Understory Structure | | | Terrain Topography | | |
|---|---|---|---|---|---|---|---|---|---|---|---|---|---|---|---|
| | n | Use | Sig. | n | Use | Sig. | n | Use | Sig. | n | Use | Sig. | n | Use | Sig. |
| Birds | 68 | 68.7 | 86.8 | 66 | 66.7 | 92.4 | 49 | 49.5 | 77.6 | 39 | 39.4 | 76.9 | 17 | 17.2 | 76.5 |
| Mammals | 23 | 69.7 | 73.9 | 21 | 63.6 | 90.5 | 16 | 48.5 | 68.8 | 11 | 33.3 | 81.8 | 9 | 27.3 | 88.9 |
| R&A | 1 | 14.3 | 100 | 2 | 28.6 | 100 | 0 | | | 0 | | | 5 | 71.4 | 100 |
| Invertebrates | 7 | 38.9 | 85.7 | 5 | 27.8 | 100 | 5 | 27.8 | 100 | 3 | 16.7 | 100 | 12 | 66.7 | 91.7 |
| BLF | 4 | 40 | 100 | 4 | 40 | 100 | 4 | 40 | 50 | 2 | 20 | 100 | 8 | 80 | 100 |
| Several | 5 | 83.3 | 80 | 3 | 50 | 100 | 4 | 66.7 | 50 | 3 | 50 | 100 | 1 | 16.7 | 0 |

**Table 3.** Categorized LiDAR-derived traits based on all metrics found in the literature search regarding vegetation structure and terrain topography, and in relation to the harmonized ecosystems. "n", "use", and "sig." values are estimated equally as in Table 2 but for the type of ecosystem.

| | Canopy Height | | | Canopy Cover | | | Canopy Vertical Distribution | | | Understory Structure | | | Terrain Topography | | |
|---|---|---|---|---|---|---|---|---|---|---|---|---|---|---|---|
| | n | Use | Sig. | n | Use | Sig. | n | Use | Sig. | n | Use | Sig. | n | Use | Sig. |
| Temperate forest | 72 | 62.1 | 84.7 | 76 | 65.5 | 93.4 | 57 | 49.1 | 77.2 | 44 | 37.9 | 75 | 30 | 25.9 | 90 |
| Rainforest | 8 | 100 | 100 | 4 | 50 | 75 | 5 | 62.5 | 60 | 0 | | | 1 | 12.5 | 100 |
| Taiga | 11 | 73.3 | 90.9 | 9 | 60 | 100 | 9 | 60 | 55.6 | 7 | 46.7 | 100 | 1 | 6.7 | 100 |
| Riparian forest | 2 | 66.7 | 100 | 1 | 33.3 | 100 | 2 | 66.7 | 100 | 1 | 33.3 | 100 | 0 | | |
| Tundra | 1 | 50 | 0 | 1 | 50 | 100 | 0 | | | 0 | | | 1 | 50 | 100 |
| Savannah | 4 | 57.1 | 25 | 3 | 42.9 | 100 | 1 | 14.3 | 100 | 1 | 14.3 | 100 | 5 | 71.4 | 60 |
| Agroecosystem | 9 | 81.8 | 88.9 | 5 | 45.5 | 80 | 4 | 36.4 | 75 | 4 | 36.4 | 100 | 5 | 45.5 | 60 |
| Arid & semi-arid ecosystem | 0 | | | 2 | 40 | 100 | 0 | | | 1 | 20 | 100 | 4 | 80 | 100 |
| Inland and coastal waters | 1 | 16.7 | 100 | 0 | | | 0 | | | 0 | | | 5 | 83.3 | 100 |

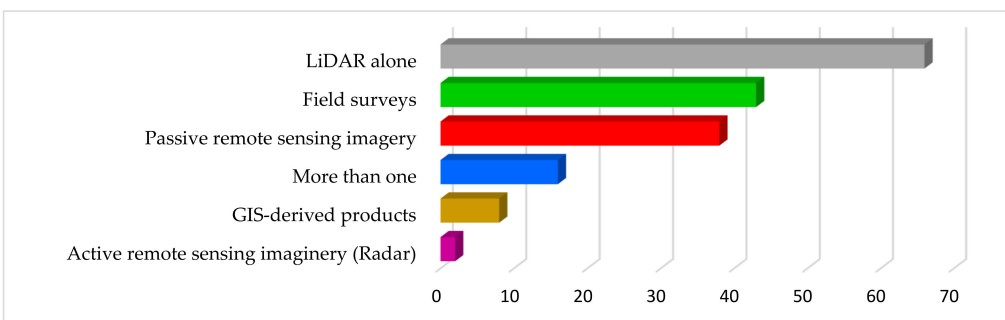

**Figure 4.** Number of articles found in the literature search using LiDAR data together with other sources: "GIS-derived products" refer to thematic maps or variables extracted from GIS analyses not linked to LiDAR data. "More than one" consider articles using at least 2 sources apart from LiDAR data, e.g., field surveys and passive remote sensing imagery.

## 4. Discussion

The application of LiDAR technology in terrestrial ecology studies has shown a consistent and expanding significance over the last two decades, clearly reflected by the upward trend in the number of articles published since 2000. Our results thus support the idea that LiDAR has become a very useful technology in animal and plant ecology studies, thanks to its key contribution in producing vegetation and terrain structure-related traits as surrogates of 3D habitat structure. We encourage researchers to increase the use of fine-grained terrain and understory and shrubland LiDAR traits, despite the latter's need for increased accuracy estimation.

We still find two main biases: First, birds are by far the most studied group, due to the recognized effect of forest structure on bird diversity (for a comprehensive review of LiDAR usefulness on bird studies, see [18]). Still, we have identified an increase of studies considering other taxonomic groups since the review of [17], including bryophytes, lichens, and fungi (see, e.g., [26,27,89]), a new broad group of organisms increasingly considered in biodiversity conservation and global change research. Studies on mammal group have also increased with successful LiDAR outcomes (see, e.g., [20,21,40,42,66,90,91]. Something similar happened with invertebrates, although to a lesser extent (see, e.g., [12,22,23]). This fits with ecological theory, which recognizes the importance of the diversity of structures as surrogates of the biodiversity itself. Regrettably, we found few studies considering reptiles and amphibians, despite the ability of LiDAR to produce accurate DTMs relevant for modelling the habitat features of small reptiles [79], the nesting sites of marine turtles according to the beach geomorphology (see, e.g., [51,92,93]), or for the identification and characterization of vernal ponds for amphibians [94]. In addition, our review retrieved few studies including several taxonomic groups (however, see, e.g., [36,49,62]). Thus, considering the pressures that are currently affecting ecosystems and biodiversity worldwide, we claim that the global studies integrating several species and taxonomic groups owe thanks to the incoming LiDAR spaceborne missions (see below).

The second acknowledged bias is the geographic distribution and the ecosystems surveyed: We have identified a clear dominance of studies in the Northern Hemisphere, particularly in the temperate forests of Europe and North America, which is consistent with previous reviews [17,18]. Studies in Central and Eastern Asia, Africa, and South America are clearly lacking. Moreover, the number of articles in rainforest, savannah, tundra, and desert ecosystems is still scarce when compared to their ecological relevance, their spatial extent, and their vulnerability to global change, despite LiDAR-derived metrics haven been proven to be good indicators of ecosystem structure. The lower availability of LiDAR data in such ecosystems and continents helps explain these geographical gaps. Space missions such as GEDI and ICESat-2, in combination with ALS and UAS LiDAR, will provide the opportunity to close these gaps, scaling up local studies to regional and global studies (however, see below).

In turn, the ability of LiDAR to accurately explain biodiversity patterns in open landscapes is evident in light of our review. This is especially so for predicting species richness in BLF [27], since these taxa are not only affected by old-growth forest structure [26,37], but also by shrub cover and terrain topography [27], i.e., traits that can be estimated from LiDAR. Indeed, terrain traits derived from LiDAR data such as slope, soil moisture, or solar radiation were also decisive for modelling biological crusts in open-arid landscapes [30]. Furthermore, LiDAR was crucial in mapping the breeding habitats of shorebirds in Oceania (*Thinornis cucullatus cucullatus*) through the characterization of unvegetated dunes and tidal reefs by means of DTMs [95]. LiDAR terrain metrics were also critical for identifying small depressional wetlands, key for pond-breeding amphibians (*Notophthalmus perstriatus*, [25]). Thus, our results have shown how LiDAR fine-grained metrics describing topography have decisively contributed to the modelling of species–habitat relationships, regardless of the taxonomic group and the ecosystem studied, despite research efforts having mainly focused on elucidating the effects of 3D vegetation structure on species. As such, we recommend researchers to fully integrate fine-grained 3D terrain traits in addition to vegetation structural traits when modelling biodiversity. Moving forward to studies in aquatic ecosystems, although not considered in this review, morphological traits such as bathymetry are further becoming successfully used for modelling the seascape structure with LiDAR, to, for example, predict the diversity and abundance of fish and corals [96], to map coral reefs [97,98], to define benthic habitat complexity for reef fish assemblages [99], or to predict coral reef fish assemblages [100].

Furthermore, ecologists and conservation biologists increasingly rely on LiDAR technology for conservation and planning strategies [101]; this includes research on invasive species (*Procambarus clarkii*, [57]), critically endangered species (*Pongo pygmaeus*, [21]), or the identification of priority areas for species conservation (ground beetles, [22]). From a conservation perspective, we consider that LiDAR represents a unique, albeit scarcely explored, technology, able to monitor how global biodiversity and ecosystem services are threatened by ecosystem alterations due to anthropogenic activities. Particularly, LiDAR can contribute by quantifying how habitat destruction, the greatest threat to biodiversity, (negatively) influences populations or communities across altered landscapes.

We detected a great variability in the terminology and calculation methods of LiDAR metrics and throughout all taxonomic groups, as highlighted in a recent study [18]. For example, canopy height can be measured by several metrics, e.g., highest vegetation return, mean height, 95th percentile canopy height, 25th percentile canopy height, etc. However, a considerable number of other metrics were also used, as, for instance, in the Gini coefficient, used to measure the degree of inequality in tree size [48], metrics to characterize snags or dead wood [31,102], or the Leaf Area Density [103]. Helpful reviews of the more useful LiDAR metrics available are provided by [104] and [18]. Therefore, we consider the challenging task of harmonizing the wide array of LiDAR metrics and calculation methods, so that the comparing of model outcomes among papers can become fruitful. We propose to simplify and harmonize LiDAR metrics to more meaningful morphological traits as follows, in line with [105]: canopy height, canopy cover, canopy vertical distribution (i.e., traits of canopy structure), also including understory and shrubland, and terrain topography traits. Despite understory and shrubland traits receiving less attention, these traits play a key role in forests and non-forest ecosystems as providers of shelter and food for different taxonomic groups. This underuse is explained by the constraints to map forest canopy (cover, height) and shrublands with low pulse density [106]. Likewise, precise terrain topography, including the seascape, is crucial for modelling ecosystem structure, as commented above, therefore it should be increasingly employed by researchers.

According to our harmonization process, canopy height was the most widely used LiDAR trait globally (e.g., [22,24,66,107,108]), because it is easy to extract from the point cloud and can be well retrieved, even with a low point density ($\leq 2$ points/m$^2$). Still, more influential were canopy cover and terrain topography, the latter being the least measured trait. Furthermore, most of the reviewed studies tended to include direct LiDAR

metrics, or easily derived variables, thus valuable information could be missing. Canopy cover traits were found to be significant for all taxonomic groups (e.g., [37,109–111]), while topography was the same for characterizing habitats for amphibians and reptiles (e.g., [25,51]), invertebrates (e.g., [52,58]), and BLF (e.g., [30,112]). Our results highlight the relevance of using understory and shrubland traits for different taxonomic groups, especially for invertebrates (e.g., [46,113]) and BLF (e.g., [27]), but also for mammals (e.g., [90,114]) and, to a lesser extent, for birds (e.g., [13,115]). These metrics were less used than canopy traits, due to the difficulty of their characterization with low spatial resolution. In any case, further efforts should be made to increase LiDAR accuracy in measuring such traits. Finally, canopy vertical distribution traits were relevant for birds (e.g., [103,107]) due to their 3D use of space, but less than canopy height (e.g., [41,84]) and cover (e.g., [109,116]). This unexpected outcome may be due to the canopy vertical distribution metrics requiring more point cloud resolution to penetrate the canopy structure to accurately describe its complexity. An alternative explanation may be that these metrics are not as standardized as canopy cover and height metrics, so that model outcomes fail to describe this trait. Interestingly, canopy vertical distribution traits were also important for invertebrates (e.g., [12,59]), even for mammals (e.g., [20,42]). All these results support the idea that environmental heterogeneity, measured through the proposed five morphological traits extracted from LiDAR, creates more niches and spatial turnover of species, favoring different habitats and thus allowing more species to coexist in accordance with the habitat heterogeneity hypothesis [3].

*From Local to Global Analyses Characterizing Terrestrial Ecosystem Structure for Biodiversity with LiDAR Data. Future Directions*

LiDAR has become a much-needed technology due to its ability to deepen our understanding of the influence of fine-scale environmental heterogeneity on biodiversity, providing precise information otherwise unfeasible. However, we found some constraints related to its resolution and data accessibility. ALS LiDAR is becoming increasingly available thanks to country-wide datasets, which is appropriate for regional scales, but the low-intermediate spatial resolution may not suit some species and ecosystems. In addition, temporal analysis is still challenging due to its low temporal resolution, which is often not suited to the timescale required (however, see, e.g., [12,87,88]). The advent of UAS LiDAR, with a temporal resolution defined by user, and an increased performance in terms of spatial resolution and better laser return penetration than ALS LiDAR, can become a good and cheaper candidate, especially at local scales [117]. UAS LiDAR will also succeed in measuring understory and shrublands, even grasslands. Although our search has not yielded any studies relating habitat structure to any of our taxonomic groups, this technology has been already satisfactorily tested (see the review by [14]). Several reasons may discourage ecologists from using UAS LiDAR: (i) Its strong dependence on GPS signals, which may not be available when flying under forest canopy [118]; (ii) The safety and privacy regulatory drawbacks for UAS flights, that are expected to be overcome soon [119]; (iii) Their autonomy limitations [120]; (iv) The not always desired experience and expertise by companies that commercialize UAS products [121]. Nonetheless, we encourage ecologists and LiDAR practitioners to incorporate this technology in their studies, especially those aiming to describe fine-grained habitat characteristics required for small mammals, steppe-land birds, arthropods, amphibians and reptiles, or BLF. Something similar happens with TLS; although scarcely used, yet according to our review, TLS will also offer ecologists new opportunities of monitoring 3D structures at a very high resolution, e.g., tree structural metrics related to branching architecture (see review by [122]).

Spaceborne LiDAR missions such as the recently launched Global Ecosystem Dynamics Investigation (GEDI) or the ICESat-2 represent an unprecedented advance in LiDAR-based ecology research, given that both missions will provide critical information for mapping habitat structure for biodiversity. The GEDI mission is the first spaceborne LiDAR designed specifically to study forest structure (canopy height, cover, and vertical distribution) and topography at regional and global scales for two years with an increased

temporal resolution and at a footprint resolution of 25 m [11]. GEDI collects data globally between 51.6° N and 51.6° S latitudes, i.e., within temperate and tropical zones (only excluding polar zones). Yet, it is not clear whether such spatial resolution is appropriate for characterizing understory traits or complex canopy structures, as recent research has shown that GEDI data are unsuitable for the identification of archaeological sites in forested environments [123]. On the other hand, the Advanced Topographic Laser Altimeter System (ATLAS) instrument onboard ICESat-2 will provide elevation data of the Earth in unprecedented detail, including coastal topography. Furthermore, the forthcoming missions NASA ISRO Synthetic Aperture Radar and the European BIOMASS missions will enable fruitful synergies with the already available information. We argue that a fusion of UAS, TLS, ALS, and spaceborne datasets will open a realm of untapped research topics regarding biodiversity and ecosystem structure in the years to come.

In the same line, a core advance in ecological studies not yet developed enough comes from the fusion of active (LiDAR, Radar) and passive remote sensing datasets, enabling the combination of structural and spectral information; for example, by coupling ICESat-2 with Landsat-8 and Sentinel-2 products [124], with potential applications in biodiversity mapping and modelling. Moreover, the study of plant functional diversity through passive remote sensing (e.g., Sentinel-2 product) is an emerging field of research [125] that, if integrated with spaceborne LiDAR (e.g., ICESat-2), will allow us to combine canopy morphological traits with plant traits of unprecedent value for the characterizing of biodiversity and ecosystems. In addition, the fusion of LiDAR data and hyperspectral imagery [61] is becoming widespread through platforms that link both types of sensors (e.g., Global Airborne Observatory, NEON Airborne Observation Platform). Further, recently launched spaceborne hyperspectral missions (e.g., PRISMA, DESIS, or GF5-AHSI) and others that are planned to launch in the next years (e.g., EnMAP, CHIME, SBG) will provide invaluable information for the better monitoring and characterizing of functional traits of vegetation [126], which, combined with the structural parameter traits from LiDAR, will provide insightful data for the mapping of biodiversity. Finally, we encourage a closer collaboration between ecologists and the remote sensing community in order to overcome the challenges arising from coupling multi-platform and multi-scale datasets [102,127]. This will contribute to the gaining of a general picture of the main drivers and mechanisms behind species diversity and ecosystems' structure, and for assessing how habitat degradation will affect global biodiversity.

## 5. Conclusions

Our results show how LiDAR has become a crucial technology in providing spatially explicit information regarding the ecosystem structure in relation to biodiversity, and especially important in face of the pressures caused by anthropogenic activities on species and ecosystems. This includes organisms poorly studied until recently, such as BLF. Furthermore, we propose to simplify and harmonize LiDAR metrics into five morphological traits: canopy height, canopy cover, canopy vertical distribution, understory and shrubland, and terrain topography. Our review also provides insightful outcomes regarding the more suitable metrics for each of the taxonomic groups that may help future studies in the selection and prioritization of LiDAR metrics. Finally, we advocate (i) for the integration of spaceborne LiDAR data with the already available ALS, UAS (drones), and TLS technology, the last two of which have been barely used until now (although they are highly promising), and (ii) the coupling of these technologies with multispectral/hyperspectral information, which will allow for the exploration and analysis of new species and ecosystems.

**Supplementary Materials:** The following are available online at https://www.mdpi.com/article/10.3390/rs13173447/s1, Table S1: Number of papers found in the literature search por the period 2000–2020 according to the keyword strings and the databases used, filtered and selected. Table S2: List of species recorded in each of the 173 articles reviewed. Annex 2: List of the reviewed articles included in the systematic review (173).

**Author Contributions:** Conceptualization, P.A.; methodology, P.A., P.L.; formal analysis, P.L., C.J.-G.; investigation, P.A., P.L., C.J.-G.; resources, P.L., C.J.-G.; data curation, P.L., C.J.-G.; writing—original draft preparation, P.A., C.J.-G.; writing—review and editing, P.A., C.J.-G.; visualization, P.A., P.L., C.J.-G.; supervision, P.A., C.J.-G. All authors have read and agreed to the published version of the manuscript.

**Funding:** This research received no external funding.

**Institutional Review Board Statement:** Not applicable.

**Informed Consent Statement:** Not applicable.

**Data Availability Statement:** The study did not report any data.

**Acknowledgments:** Authors would like to thank the REMEDINAL TE-CM project (S2018/EMT-4338), and three anonymous reviewers that helped improve the manuscript.

**Conflicts of Interest:** The authors declare no conflict of interest.

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
