# Peer review of "Disentangling LiDAR Contribution in Modelling Species–Habitat Structure Relationships in Terrestrial Ecosystems Worldwide. A Systematic Review and Future Directions"

_remotesensing, doi:10.3390/rs13173447_

Round 1

Reviewer 1 Report

The majority of edits strengthened the paper with one exception, the additions to the abstract.  Line 15 and the final two sentences of the abstract need to be revised for clarity.  

Line 15.  We quantify where, in which ecosystems, what kind of LiDAR data current studies make use of LiDAR metrics, the platforms used and LiDAR data availability.

We quantify the types of research (ecosystem and LiDAR sources) and describe the LiDAR platforms and data that are currently available.

Line 25.

Finally, we discuss the forthcoming opportunities for biodiversity mapping with different LiDAR platforms and in combination with spectral information emerging from recent and future platforms and missions. We claim (i) for the integration of spaceborne LiDAR data with the already available airborne (airplane, drones) and terrestrial technology, and (ii) it coupling with multispectral/hyperspectral information, that will further let to explore
and analyze new species and ecosystems, bringing the gaps that yet exist.

Finally, we discuss the forthcoming opportunities for biodiversity mapping with different LiDAR platforms and in combination with spectral information. We advocate (i) for the integration of spaceborne LiDAR data with the already available airborne (airplane, drones) and terrestrial technology, and (ii) coupling it with multispectral/hyperspectral information will allow for the exploration and analyses for new species and ecosystems.

Author Response

Comments and Suggestions for Authors

The majority of edits strengthened the paper with one exception, the additions to the abstract.  Line 15 and the final two sentences of the abstract need to be revised for clarity.  

Response to reviewer: Thank you for your comments and for thinking that this new version of our manuscript has improved considering the deep review performed previously. Of course, we have included both of your suggestions in the abstract.

Line 15.  We quantify where, in which ecosystems, what kind of LiDAR data current studies make use of LiDAR metrics, the platforms used and LiDAR data availability.

We quantify the types of research (ecosystem and LiDAR sources) and describe the LiDAR platforms and data that are currently available.

Response to reviewer: Done (replaced)

Line 25.

Finally, we discuss the forthcoming opportunities for biodiversity mapping with different LiDAR platforms and in combination with spectral information emerging from recent and future platforms and missions. We claim (i) for the integration of spaceborne LiDAR data with the already available airborne (airplane, drones) and terrestrial technology, and (ii) it coupling with multispectral/hyperspectral information, that will further let to explore
and analyze new species and ecosystems, bringing the gaps that yet exist.

Finally, we discuss the forthcoming opportunities for biodiversity mapping with different LiDAR platforms and in combination with spectral information. We advocate (i) for the integration of spaceborne LiDAR data with the already available airborne (airplane, drones) and terrestrial technology, and (ii) coupling it with multispectral/hyperspectral information will allow for the exploration and analyses for new species and ecosystems.

Response to reviewer: Done (replaced)

Reviewer 2 Report

The article is very interesting, considering that there are only 173 articles in this field, it is important to encourage researchers to use a technology that seems to work in the study of ecosystems, especially those at risk.

Review:

- The table 2 and 3 could be improved, especially in the style.

- Figure 4 is not very explicative, it could be replaced with another one that is clearer and visibly better.

- Why does the reference text take up more space?

Author Response

Comments and Suggestions for Authors

The article is very interesting, considering that there are only 173 articles in this field, it is important to encourage researchers to use a technology that seems to work in the study of ecosystems, especially those at risk.

Response to reviewer: Thank you very much for your kind and constructive comments.

 Review:

- The table 2 and 3 could be improved, especially in the style.

Response to reviewer: we have tried to improve the style of both tables, despite we are not sure what the reviewer is expecting without providing further information.

- Figure 4 is not very explicative, it could be replaced with another one that is clearer and visibly better.

Response to reviewer: thank you for your suggestion. We have replaced it by a new one which is clearer and more informative.

- Why does the reference text take up more space?

Response to reviewer: We are not sure about this comment. Do you mean that the references in the reference list are wider than the rest of the text? We have formatted the manuscript according to the Microsoft Word Template provided by the Journal.

Reviewer 3 Report

General

Overall, this is a very interesting and detailed review on the use of LiDAR in terrestrial ecosystems. It is a shame that its use in marine systems was excluded, which might alter the percentages of use across countries, particularly Australia where there is a major use of this system. It is worth justifying this exclusion and amending the title to clarify your focus on terrestrial systems only. Please note, the English language and tense use throughout require careful checking and revision. Once these changes have been made, this will make a useful contribution to the manuscript. Major revisions.

Line 27, this phrasing/sentence is unclear – it is not clear how what you claim fits with the points – the sentence does not make sense.

Line 191, tense use should be past for the results section; please check this is consistent throughout. Here is should be was.

Methods – general question, why were marine animals left out – there are many studies on marine life using LiDAR? This exemption should be justified. The title should clarify terrestrial only. Also there are some studies on sea turtle tracks on nesting beaches using LiDAR – why have these been excluded?

Line 324, sentence incomplete, clarify the % and add studies after both percentages.

Line 324, data are plural, please amend

Line 369, several studies have been conducted exploring beach habitat used by marine turtles; please check these out.

Author Response

Comments and Suggestions for Authors

General

Overall, this is a very interesting and detailed review on the use of LiDAR in terrestrial ecosystems. It is a shame that its use in marine systems was excluded, which might alter the percentages of use across countries, particularly Australia where there is a major use of this system. It is worth justifying this exclusion and amending the title to clarify your focus on terrestrial systems only. Please note, the English language and tense use throughout require careful checking and revision. Once these changes have been made, this will make a useful contribution to the manuscript. Major revisions.

Response to reviewer: Thank you very much for your kind and helpful comments and suggestions. We really appreciate your comment regarding the lack of information about marine systems, but our manuscript focused on terrestrial ecosystems, despite we included several papers considering coastal or inland water systems. This is in part due to our review firstly focused on fauna, and then where the studies were conducted (type of ecosystem): vertebrate (birds, mammals, reptiles and amphibians) and invertebrate fauna (only species belonging to arthropod and mollusk phyla were registered), and on bryophytes, lichens and fungi. Consequently, fishes were not incorporated, and thus marine ecosystems were not included either, except in particular cases, such as the article by Culver et al 2020 (Frontiers in Marine Science) on nesting sites and habitat selection by the sea turtle Lepidochelys kempii. However, we included in the discussion a reference to the LiDAR relevance for modelling seascapes “As such we recommend to fully integrate fine-grained 3D terrain traits in addition to vegetation structural traits when modelling biodiversity, moving forward to studies in aquatic ecosystems, where these morphological traits are further beginning to be used for modelling seascapes e.g. in predicting coral reef fish assemblage structure (see e.g. 94 (Weeding et al. 2019)).” In any case, we have expanded this information in the new version, pointing to the current and potential contribution of LiDAR in marine ecosystems. Indeed, there are several Special Issues currently open in Remote Sensing journal such as Remote Sensing Applied to Marine Species Distribution, Monitoring Aquatic Environments Using LiDAR or even the Special Issue Monitoring Aquatic Environments Using LiDAR that may complement our manuscript, since they specifically include LiDAR bathymetry as a keyword. For all this, we believe that it is more appropriate to specify that our review considers terrestrial ecosystems, and, consequently, we have included it in the title, following your suggestion.

Regarding the English language and tense use, we have carefully checked and corrected them.

 Line 27, this phrasing/sentence is unclear – it is not clear how what you claim fits with the points – the sentence does not make sense.

Response to reviewer: Rephrased according to the reviewer 1’s suggestion.

Line 191, tense use should be past for the results section; please check this is consistent throughout. Here is should be was.

Response to reviewer: Done

Methods – general question, why were marine animals left out – there are many studies on marine life using LiDAR? This exemption should be justified. The title should clarify terrestrial only. Also there are some studies on sea turtle tracks on nesting beaches using LiDAR – why have these been excluded?

Response to reviewer: As we mentioned above, we did not deliberately include in our queries “LiDAR and Fish”, and for that reason we left out this group in the review. Regarding the studies on sea turtles, we have carefully checked and reviewed our selection criteria (“LiDAR AND Reptiles”, “LiDAR AND Vertebrates”, “LiDAR AND Biodiversity”), and the only paper retrieved was the one already included:

Culver, M.; Gibeaut, J.C.; Shaver, D.J.; Tissot, P.; Starek, M. Using Lidar Data to Assess the Relationship Between Beach Geomorphology and Kemp’s Ridley (Lepidochelys kempii) Nest Site Selection Along Padre Island, TX, United States. Front Mar Sci 2020, 7, 214, DOI 10.3389/fmars.2020.00214.

Anyhow, we have conducted a new search using the keyword string “LiDAR AND Turtles” and we have found five articles using LiDAR for the study of nesting sites by sea turtles in beaches, with similar outcomes as the one by Culver et al. 2020.  As all of them have in common that LiDAR provide precise information regarding the beach topography for turtles’ nesting sites, we have expanded this information in the discussion.

Line 324, sentence incomplete, clarify the % and add studies after both percentages.

Response to reviewer: Done

Line 324, data are plural, please amend

Respond to reviewer: Done

Line 369, several studies have been conducted exploring beach habitat used by marine turtles; please check these out.

Response to reviewer: Done. We have included a couple of more references apart from the one we already had.

Round 2

Reviewer 3 Report

The authors have addressed all reviewer comments adequately.

This manuscript is a resubmission of an earlier submission. The following is a list of the peer review reports and author responses from that submission.

Round 1

Reviewer 1 Report

The paper focuses on the review paper"Disentangling LiDAR contribution in modelling species-habitat structure relationship worldwide. A systematic review and future directions". The study identifies the use of LIDAR metrics in ecological applications regarding various species. I am happy with the review of LiDAR metrics used in ecological studies. However, I am also interested in the review of specific modelling techniques that are used to link LiDAR metrics to ecological parameters such as diversity. For example, the use of parametric and no-parametric empirical models....If I missed them, please highlight them.

Reviewer 2 Report

I have reviewed the paper “Disentangling LiDAR contribution in modelling species-habitat structure relationships worldwide. A systematic review and future directions” by Acebes, Lillo and Jaime-González. The paper aims to review the use of LiDAR to analyze 3D habitat structure (vegetation physiognomy and topography) in relation to abundance, occurrence, richness, and diversity of species, including behavioral studies. The paper is well written. Methods used well described and the results celery presented. I was kind of excited about this review, as I was very curious to about the synthesized knowledge on how LiDAR methods can contribute to a better understanding of species-habitat structure relationships (as indicated by the title). Unfortunately, the paper do not deliver this, and I ended up rather disappointed after reading the paper. Rather than providing a synthesis of the results and knowledge generated by the papers essentially just summarize the number of studies utilizing LiDAR data on different species groups and habitat types. The results as presented was in this context a clear disappointment. I was, however, still a bit optimistic when I saw the heading “From local to global analyses characterizing habitat structure for biodiversity with LiDAR data. Future directions”, in the discussion. Unfortunately, this section was also a disappointment, as it is not at all is based on a synthesis of the reviewed paper, and in my opinion, essentially could have been formulated without reviewing the 173 papers. In a way this highlights what find is the major problem/weakness of the paper, that there is no real synthesized results presented. In my opinion, a paper that just tally up the number of LiDAR papers published and summarize the number of studies that covered different species groups of habitats is of rather limited value. I expect much more than this from a modern review paper. I had expected some type of synthesis of results that advanced the knowledge front, e.g. by highlighting under what circumstances LiDAR might be a useful method and when it has been proved not useful. The part of the discussion from L. 383 to 395 is, in this context a bit of an exception and clearly closer to what I had expected. I find it a bit of a pity that the paper cannot get any further than tis given the time and effort that has been put into scrutinizing 173 papers.

The work and the conclusions made almost appears a bit superficial. This makes my disagree with one of the major conclusions “Our results show how LiDAR technology has greatly contributed to habitat mapping, including 458 organisms such as birds and mammals…”. Since the authors never presents any results that actually demonstrates HOW this technology have contributed, they essentially just report that there are studies that have addressed these issues, the conclusion is a bit overstating.

Specific comments

L. 31-32. Ok, but would be better if this could be exemplified. For example in what way has LiDAR techology led to a great improved understanding of the habitat use for poorly studied species? Rather than just saying poorly studied species, specify, which the of the studied species groups that you refers to here.

L. 44. Structures are not only seen as determinant of habitat quality, but also as an indicator of biodiversity itself, i.e. structural diversity is commonly used as a surrogate for species diversity.

In the aim, the authors state that “We focus on non-avian studies, due to the recent review performed on birds [18]” but still bird studies are included the survey, and take up a large part of the descriptive results presented, why?

L. 291-291: The text is strange here “…between two and ten points/m2 were recorded in 24.3% (N=42), whereas 15.6% of the articles had more than 10 points/m2 in (N=27). One study used less than 2 points/m2 and more than 10 [87].” As a reader, you first get the impression that 15.6% of the studies used more than 10 p /m2, but the following sentence can be read like there was one study with less than 2 and one with more than 10. The more than 10 part is confusing.

L. 336. “Our review also provides insightful results regarding the more suitable metrics for each of the taxonomic groups.” In my mind, it would be much more interesting if you explained and exemplified what these insights are rather than just saying that they exist.

Reviewer 3 Report

  1. "LiDAR" or "LIDAR", be sure of it.
  2. Literature review of LiDAR for forest structure and biomass estimation need to enhanced.
  3. Review for the current popular LiDAR instruments and their characters is recommended.
  4. Future direction part need to enhanced, e.g., fusion of LiDAR and others remote sensing datasets for more advanced biodiversity studies, potential of multispectral and hyperspectral LiDAR instruments.

Reviewer 4 Report

This is a well written literature review exploring the recent applications of LiDAR technology to modeling species-habitat structure.  The authors address a diverse set of taxa including vertebrates, bryophytes, lichens, fungi and invertebrates.  In addition, the summarize the trends through time, and geographic disparities is use of the technology.  Finally, they point out the importance of publicly available data in driving innovation.

There are only a few minor edits required for clarity. 

Line 26    Globally, canopy height and canopy cover were the most used LiDAR metrics, especially in forest ecosystems, whereas canopy cover and terrain topography metrics showed higher performance in those ecosystems that were mapped.

…metrics performed better, were found to be significant, in those ecosystems where they were mapped.

Line 142.   This index was estimated as the percentage of articles where the metrics resulted significant.

This index was estimated as the percentage of articles where the metrics were identified as significant.

Line 193  Reptiles and amphibians are the less represented group (N=7, 4%).

..Reptiles and amphibians are the least represented groups (N=7, 4%)

Line 194  - BLF is the only one where studies have been conducted in North America and Europe, particularly in latitudes above 48ºN (Alaska, Canada, Denmark, Finland, Germany and  Norway), except one study predicting biological crust in arid environments in Southern [30].

… arid environments in the Southern ?  [30]. (Hemisphere?)

Line 225 Temperate forest is by far the most represented ecosystem (N=116, Figure 3a) and the most common across all taxonomic groups (Figure 3b). Contrarily, riparian forests (only bird studies), savannah (mammals and invertebrates) and tundra ecosystems (birds and BLF) were the less  represented (Figure 3b).

… Riparian forests (only bird studies), savannah (mammals and invertebrates) and tundra ecosystems (birds and BLF) were the least represented (Figure 3b).

Line 248 Globally, canopy cover and topography metrics showed the highest performance (93.2% and 86.5%, respectively), followed by canopy height (84.3%), understory structure and shrublands (81.0%), and canopy vertical distribution (74.4%).

Globally, canopy cover and topography metrics performed best (93.2% and 86.5% respectively), …..

Line 251 By taxonomic group, canopy height was the most used LiDAR metric in birds and mammals, despite canopy cover was more influential (Table 2).

By taxonomic group, canopy height was the most used LiDAR metric in bird and mammal studies, even though canopy cover was more influential (Table 2).

Line 252 Although relevant, canopy vertical distribution metrics were less important for birds than canopy high and cover.

Although relevant, canopy vertical distribution metrics were less important for birds than canopy height and cover.

Line 254 Understory and shrublands metrics were less used than other vegetation structure metrics in BLF and invertebrates, despite they were more influential for those taxa.

Understory and shrublands metrics were used less often than other vegetation structure metrics in BLF and invertebrate studies, despite the fact they were influential for those taxa

Line 282 Seven out of 164 articles mounted the LiDAR sensor on a helicopter instead of on an airplane [12,32,81-85], increasing LiDAR due to helicopter’s flight is lower and slower resolution (pulse density: 15-500 pulses/m2).

Unclear what this sentence is trying to communicate …

Line 333. Indeed, ecologists and conservation biologists increasingly rely on LiDAR technology for conservation and planning strategies [88]; this includes research on invasive species (e.g., Procambarus clarkii, [57]), critically endangered species (e.g., Pongo pygmaeus, [21]) or the identification of priority areas for species conservation (e.g., ground beetles, [22]).

Pongo pygrmeaus – should be italized

Line 376  African savannah harbors recent studies such as the effects of termite mounds in shaping landscape heterogeneity [28], the effects of vegetation structure on lion Panthera leo kill sites [67] or habitat preferences of sable antelope Hippotragus niger in relation to landscape structure [69].

African savannahs have been the focus of recent studies, such as quantifitying the effects of termite mounds in shaping landscape heterogeneity [28], the effects of vegetation structure on lion (Panthera leo) kill sites [67] or habitat preferences of sable antelope (Hippotragus niger) in relation to landscape structure [69].

Line 396 We detected a great variability in the terminology and calculation methods of LiDAR metrics and throughout all taxonomic groups, in line with the outcome of [18].

Unclear what the last phrase of this sentence is trying to communicate, incomplete sentence?

Line 405 Canopy cover metrics showed great performance for all taxonomic groups (e.g. [37,103- 105]) , while topography was so for amphibians and reptiles (e.g., [25,51]), BLF (e.g.,[30,106]) and invertebrates (e.g., [52,58]).

Canopy cover metrics were found to be significant for all taxonomic groups (e.g. [37,103- 105]) , while topography was found to be significant for characterizing habitat for amphibians and reptiles (e.g., [25,51]), BLF (e.g.,[30,106]) and invertebrates (e.g., [52,58]).

Line 438   Nonetheless, we deeply encourage ecologists to incorporate this technology in their studies, especially for those aiming to describe fine-grained habitat characteristics required for small mammals, step-land birds, arthropods, amphibians and reptiles or BLF.

Nonetheless, we encourage ecologists to incorporate this technology in their studies, especially those aiming to describe fine-grained habitat characteristics required for small mammals, steppe-land birds, arthropods, amphibians and reptiles or BLF.